# Chronic Undernutrition in Ovine Twin Pregnancies Abolishes Differences in Birth Weight Due to Sex: An Evaluation of the Role of Nutritional and Antioxidant Supplementation

**DOI:** 10.3390/ani14060974

**Published:** 2024-03-21

**Authors:** Francisco Sales, Óscar A. Peralta, Mónica De los Reyes, Camila Sandoval, Paula Martínez-Ros, Carolina Rojas, Antonio Gonzáles-Bulnes, Víctor H. Parraguez

**Affiliations:** 1Instituto de Investigaciones Agropecuarias (INIA), CRI-Kampenaike, Punta Arenas 6212707, Chile; fsales@inia.cl (F.S.); camila.sandoval.torres@inia.cl (C.S.); 2Escuela de Medicina Veterinaria, Facultad de Agronomía e Ingeniería Forestal, Facultad de Ciencias Bioloógicas y Facultad de Medicina, Pontificia Universidad Católica de Chile, Santiago 7820436, Chile; oscar.peralta@uc.cl; 3Facultad de Ciencias Veterinarias y Pecuarias, Universidad de Chile, Santa Rosa 11735, Santiago 8820808, Chile; mdlreyes@uchile.cl (M.D.l.R.); carolina.rojas.t@ug.uchile.cl (C.R.); 4Facultad de Veterinaria, Universidad CEU Cardenal Herrera, CEU Universities, C/Tirant lo Blanc, Alfara del Patriarca, 46115 Valencia, Spain; paula.martinez@uchceu.es (P.M.-R.); antonio.gonzalezbulnes@uchceu.es (A.G.-B.); 5Facultad de Ciencias Agronómicas, Universidad de Chile, Santa Rosa 11315, Santiago 8820808, Chile

**Keywords:** sheep pregnancy, fetal sex, twins, nutrition, antioxidants

## Abstract

**Simple Summary:**

The fetal growth pattern in twin pregnancies is usually affected by the sex of the co-twin pairs, among other factors, with females being lighter than males at birth. We aimed to determine the effect of the sex of co-twins on lambs’ birth weight in ovine pregnancies developed under natural undernourishment, a condition of sheep farming frequently encountered worldwide. Additionally, we sought to investigate whether the nutritional and/or antioxidant supplementation provided to ewes during pregnancy could modulate the potential effects associated with the sex of co-twins. We found that maternal nutrient restriction abolishes the sex differences in body weight at birth between co-twins. However, improving maternal nutrition and supplementation with antioxidants during gestation not only improves maternal weight and body condition but also tends to restore birth weight and its differences between female and male offspring, an effect that is enhanced with the combined supplementation of concentrated food and antioxidants. These results suggest that undernutrition not only may alter the intrauterine growth through the maternal–fetal relationship, but also through the feto–fetal relationship, which can be normalized via strategically targeting interventions such as maternal nutrient supplementation or antioxidant supplementation during gestation. Thus, a better understanding of the driving factors of this effect is of major relevance to improving fetal growth and lamb survival in harsh environments.

**Abstract:**

In twin pregnancies of discordant sex, the male fetus grows larger than the female co-twin. Our study aimed to determine the effect of the sex of co-twins on lambs’ birth weight in ovine pregnancies developed under natural undernourishment. Additionally, we investigated whether the nutritional and/or antioxidant supplementation provided to ewes during pregnancy could modulate the potential effects associated with the sex of co-twins. Ninety-six birth records of twin pregnancies of sheep grazing the natural Patagonian prairies were analyzed. The animals were divided into four groups: control (no supplementation), N (concentrate supplementation, 100% NRC), A (antioxidant supplementation), and NA (concentrate + antioxidant supplementation). Supplementation occurred from day 35 of gestation onwards until lambing. There were no differences in female or male birth weight in the control undernourished group. However, in group N, females or males with sex-discordant co-twins had a higher birth weight than did those with co-twins of the same sex. Group A males with female co-twins had a higher birth weight compared to males whose co-twins were also males. In NA lambs, males had a higher birth weight compared to females, regardless of their co-twin’s sex. Therefore, chronic undernutrition abolished the differences in birth weight due to fetal sex. Restoring maternal nutrition or antioxidant supplementation tends to normalize birth weight and restore the differences between females and males. This effect is enhanced with the combined supplementation of concentrated food and antioxidants.

## 1. Introduction

The discordant fetal growth and birth weight between males and females in twin pregnancies has been largely recognized in humans, with male newborns being heavier than females [1,2,3]. Existing data indicate that the sex of each fetus in twin pairs affects the growth of the co-twin and, overall, male–male twins are heavier at birth than male–female and female–female twin pairs. In the case of within-sex-discordant twin pairs (male and female), males are also heavier than females [3].

Similar observations extend to twin pregnancies in sheep, where male lambs’ birth weight is about 5–10% higher than that of female littermates [4,5,6,7]. Female birth weight is more reduced when the co-twin is a male than in the case of a female co-twin; conversely, the body weight reduction in male twins is not related to their co-twin’s sex [6].

In current sheep production, there is a growing demand from sheep farmers to improve efficiency on a global scale. Increasing the number of descendants per sheep and pregnancy (i.e., increasing prolificacy) has become an important breeding goal to improve not only efficiency but also productivity, as well as to promote the economic value of sheep farming [8,9]. Consistent with the above, there are intense efforts to introduce prolific breeds to improve productivity, both in extensive [10] and intensive [8] sheep breeding systems. Consequently, there is a significant increase in twin pregnancies which, on the other hand, may negatively affect efficiency due to an increased risk of perinatal mortality because of the low birth weight of lambs in twin pregnancies [11,12].

The occurrence of low birth weight and related events in twin pregnancies increases in the case of maternal undernutrition. Such an event has been largely studied in humans, in which the metabolic rate of mothers with multiple pregnancies has been found to be 10% greater than that of women with singleton pregnancies [13]. Two fetuses require a continuous supply of nutrients for growth [14] and, therefore, the need for adequate maternal nutrition increases to prevent the likelihood of poor fetal growth [15,16]. This is a concerning problem in sheep production since worldwide sheep farming mainly takes place in agriculturally marginal territories, with little availability of water and pasture [17,18]. Chilean Patagonia is a good example of these territories, where sheep breeding mainly takes place under extensive natural pasture-based systems, which offer limited environmental and nutritional conditions to pregnant ewes [19]. According to the NRC requirements for twin-bearing ewes [20], the pasture nutrient allowance under Patagonian conditions would cover about 72% of their energy requirements but only 30% of their protein requirements during wintertime. Such undernutrition results in a marked reduction in maternal body condition throughout pregnancy, leading to fetal growth restriction and low birth weight [21,22].

Our studies on undernourished sheep twin pregnancies reared in natural prairies in Magellan Patagonia have not found significant effects of sex on fetal or newborn body weight [21,22,23]. However, the distribution of sex in these studies is random. Hence, in the present work, we specifically aimed to determine the effect of co-twin sex on lamb’s birth weight in ovine pregnancies developed under natural undernourishment.

Fetal growth restriction involves not only inadequate nutrient supply but also reduced oxygenation, which causes hypoxia that increases oxidative stress and triggers low-grade inflammation [24]. The antioxidant defense system is weakened, thus exacerbating oxidative stress [25,26,27]. Our prior studies in sheep have shown that the effects of maternal undernutrition and subsequent fetal growth restriction are associated with fetal hypoxemia and oxidative stress [21]. Hence, the effects of undernutrition and increased oxidative stress can be counteracted by supplementing sheep with concentrated feed, antioxidants, or a combination of both during pregnancy [23,28,29]. In this study, we aimed to investigate whether nutritional and/or antioxidant supplementation, which was provided to ewes during pregnancy, could modulate the potential effects associated with the sex of co-twins.

## 2. Materials and Methods

### 2.1. Animals and Experimental Procedure

Ninety-six birth records of twin sheep pregnancies from the experimental flock of the Kampenaike experimental station (Instituto de Investigaciones Agropecuarias, INIA, Magellan Region, Chilean Patagonia; latitude of 52°36′ and longitude of 70°56′) were used. These records were obtained from other trials conducted in the experimental station during the same breeding season under standard commercial breeding conditions and involved twin-bearing Corriedale ewes with normal deliveries and live newborn lambs that suckled normally. The average body weight (BW) and body condition score (BCS; scale 1–5 [30]) of the animals at the beginning and at the end of pregnancy were 60.9 ± 0.6–2.3 ± 0.1 and 66.1 ± 1.1–1.6 ± 0.1, respectively. The individual records of the lambs included the weight and sex (female: F; male: M) of each twin obtained before 12 h from parturition. The newborn lambs’ weight was measured using a digital hand scale with a sensitivity of 5 g in the range of 0–5 Kg (Generic Scale^®^, Mumbai, India).

Nutritional status and treatment were the same in all trials. In brief, all pregnant ewes were allowed to graze freely during the entire gestation period on a natural prairie composed mainly of *Festuca gracillima* and *Chiliotrichium diffusum* (total dietary nutrients of about 45%, with 3.3% crude protein and 1.9 Mcal/kg metabolizable energy). The stocking rate was 0.9 ewes per hectare and the availability of dry matter was 255 kg per hectare, which is representative of southern Patagonian prairie conditions. Under these foraging conditions, twin-bearing ewes are unable to meet their nutritional requirements, especially during the last third of pregnancy (PC: 165 g/day, EM: 3.5 Mcal/day [31]); therefore, a state of undernutrition was developed because the ewes did not reach the expected weight gain during gestation at a level that was at least similar to that of the conceptus mass [32]. On day 35 after mating, the ewes were submitted to ultrasound scanning via an abdominal approach to confirm twin pregnancy. Four groups of twin-bearing ewes were included in this study, as described in Table 1.

Supplementation with concentrate food and/or antioxidants was given from day 35 of gestation onwards until lambing. The N group, with nutritional supplementation, received a daily administration of 450 g of concentrate feedstuff (17.0% crude protein; 3.0 Mcal/kg metabolizable energy) per animal. For the A group, the antioxidant supplementation was given in the form of herbal-based products containing polyphenols, whose antioxidant activity is similar to that of vitamins C and E (C-Power^®^ 10 g kg^−1^ and Herbal-E50^®^ 7 g kg^−1^; Nuproxa, Etoy, Switzerland); this premix was in a concentrate of similar characteristics to that supplied to the N group. Thus, each pregnant ewe in the antioxidant-treated groups received 50 g of this premix containing a total dose of 500 mg of C-Power^®^ and 350 mg of Herbal E50^®^ daily, which has previously been shown to increase maternal and fetal antioxidant capacity [28]. The supplementation for the NA group consisted of daily administration of 400 g of concentrate without antioxidants plus 50 g of the premix with antioxidants, while in the C group, the animals received daily administration of 50 g of concentrate without antioxidants.

### 2.2. Statistical Analysis

The experimental model was a 3 × 2 factorial design, where the newborn sex (female or male), feeding regime (grazing or grazing plus concentrate), and antioxidant supplementation (with or without antioxidant supplementation) were the fixed effects. The lambs’ birth weight was compared by means of the analysis of variance, using the general linear model procedure of SAS (GLM; SAS Institute Inc., Cary, NC, USA) after normality testing of the data. Maternal BW recorded at the end of the trial was considered a covariate. In addition, the ewes’ BW and BCS were compared at the beginning and the end of pregnancy in each experimental group using Student’s *t*-test. Differences were considered significant when *p* ≤ 0.05, while the results were considered to show only a tendency when the *p*-value was between 0.05 and 0.10. The results are presented as mean ± SEM.

## 3. Results

### 3.1. Ewes’ BW and BCS during Pregnancy

Maternal body weight (BW) and body condition score (BCS) are shown in Table 2. The initial BW and BCS were similar for all groups. At the end of gestation, the groups that received the concentrate (N and NA) significantly increased their BW but showed no change in their BCS. In contrast, groups C and A did not show a change in their BW when compared to the initial weight, but they significantly reduced their BCS.

### 3.2. Fetal Sex Pair Distribution and Birth Weight

In the four experimental groups, the frequency of FM (female–male) twin pairs tended to be double that of FF and MM (female–female and male–male, respectively) pairs. Likewise, FF twin pairs were more abundant than MM twin pairs, with the exception of the control group in which there were more MM than FF twin pairs (Table 3).

Considering all female and male newborns in each group, birth weight was similar between the sexes in the control undernourished group (C). In the groups supplemented with either concentrate (N) or concentrate plus antioxidants (NA), males were significantly heavier than females, whilst in the group supplemented with antioxidants (A), males showed a trend of being heavier than females.

### 3.3. Effects of Supplementation with Concentrate and/or Antioxidants on Lambs’ Birth Weight in Accordance with Their Co-Twin Sex

Table 4 shows the effects of maternal nutritional and/or antioxidant supplementation on the birth weight of male and female twin lambs and their sex-concordant or discordant co-twins.

In females with male co-twins (F(M)), maternal supplementation with concentrate feed, either alone or in combination with antioxidants, had a significant effect on increasing birth weight by about 19–20%, compared to the controls. No effect was observed with supplementation with only antioxidants.

In females with female co-twins, birth weight was positively affected by concentrate and antioxidant supplementation, although individually administering each supplementation by itself did not achieve statistical significance. In this case, the treatment that achieved the greatest effect on birth weight in female lambs with a female co-twin was the supplementation with concentrate plus antioxidants (group NA), with an increase of 21.0% above the controls.

In males with female co-twins, birth weight was significantly increased by maternal supplementation with concentrate feed or antioxidants independently, as well as in combination. The greatest increase in birth weight (32.9%) was observed in the group treated with concentrate feed (N), followed by the group whose mothers received concentrate feed plus antioxidants (NA), with a weight increase of 30.3% compared to the controls.

In males with male co-twins, birth weight was positively affected by maternal supplementation with concentrate feed and antioxidants, showing a significant effect when both kinds of supplementation were administered simultaneously (an increase of 40.3% above the controls).

When analyzing the relationships between fetal co-twin sexes and experimental groups, it was observed that there were no differences in female or male birth weight in the control undernourished group (C), regardless of whether the co-twins were of concordant or discordant sex. The newborn lambs from the ewes supplemented with concentrate feed (group N) showed an effect of co-twin sex on birth weight, wherein females or males with sex-discordant co-twins had a higher birth weight than those with co-twins of the same sex. In this group, the newborn males with female co-twins had the highest birth weight. When the ewes were supplemented with antioxidants (group A), male lambs with co-twins of discordant sex showed the highest weight at birth, followed by males or females with co-twins of the same sex. There was no effect of maternal supplementation with antioxidants on the birth weight of female newborns with a male co-twin. In the group whose mothers received a combination of concentrated feed and antioxidants (group NA), male newborn lambs with male co-twins showed the highest birth weight, followed by male lambs in female–male couples. No differences in female body weight were observed, regardless of whether the co-twins were of concordant or discordant sex. It is important to note that in the three treatments, where twins were of discordant sex, the birth weights of female lambs were significantly lower than those of male lambs (Table 4).

## 4. Discussion

The present results support that maternal undernutrition and subsequent nutritional supplementation during pregnancy have a sex-related differential effect on offspring development. It is widely known that, under adequate environmental and nutritional traits, lamb birth weight is higher in males than females [4,5,6,7]. Our data indicate that birth weight in undernourished twin pregnancies is more unfavorable for male lambs, lowering their weight to a similar value as their female counterparts. These results are in agreement with our previous studies under the same condition [21,22,23]. In the present study, such an effect was counteracted in the groups supplemented with either concentrate feed or antioxidants, as well as the group supplemented with a combination of both. Hence, males remained heavier than females, suggesting that undernutrition induces a stronger effect on the development of male than female offspring. Therefore, our results support evidence of sex-specific responses to fetal programming interventions [33].

Evidence of sex-specific responses in fetal programming has opened a new field of research since the basis for this sexual dimorphism in programming embryo/fetal development has not yet been fully clarified. It is generally believed that sex-specific differences in developmental programming models are the results of the different abilities of male and female fetuses to respond to a given stress [33]. However, there is an alternative hypothesis suggesting that all fetuses respond identically to the same insult, but maternal physiology modulates the development and performance of fetuses in a different way depending on the sex of the offspring [33]. In this sense, the Trivers–Willard hypothesis postulates that the external environment is signaled to the conceptus by the mother via the uterine environment, and such signaling influences the phenotype of the offspring produced to maximize the chances of species survival. The placenta, being a nutrient sensor that regulates nutrient transport, adapts to the ability of the mother to deliver nutrients, and, by directly regulating fetal nutrient supply and fetal growth, the placenta plays a central role in fetal programming [34]. However, both hypotheses may coexist since maternal signaling could be related to differences between the sexes in their response to maternal corticosteroids and sex steroids and in the proper secretion of steroids by the offspring [35,36].

Gestation is a state of equilibrium between the necessities of a mother and those of her progeny, and mothers do not invest more than strictly indispensable in their offspring’s growth, as a preventive strategy to ensure their own energetic balance during and after pregnancy [37,38]. In evolutionary biology, males are seen as the “most expensive and less useful” sex. Thus, it may not be energy efficient for a mother to heavily invest in male offspring when the investment in the adequate development and homeostasis of female progeny would guarantee the chance to reproduce and maintain the species through gene transmission to future generations [39,40]. Adaptations in fetal somatic tissues to environmental stresses that are costly in terms of energy investment would, therefore, be preferentially directed to female fetuses, whose physiology and reproductive functionality must be protected to ensure the possibility of reproducing and carrying out pregnancy and lactation, while males only need to protect their germ cells and be able to mate.

These sex-related differences in programming have previously been reported in murine models, in which males show a lack of protection [41,42] as the somatic development of male fetuses is sacrificed to safeguard maternal resources. In female fetuses, on the other hand, the placenta seems able to adapt more readily to adverse conditions [43,44], and this reflects the attempt to buffer and adapt to stressors in order to preserve the conceptus in excellent conditions and maximize the chances of pregnancy and raising offspring. Subsequent studies in large mammals (pigs) showed stronger protective strategies for suitable growth and postnatal survival in female conceptuses under maternal undernutrition [45,46,47]. In summary, both male and female fetuses develop adaptive processes, but female fetuses are able to maintain a better development of key viscera (brain, liver, intestine, and kidneys) at the expense of body development. Furthermore, these morphometric differences are reinforced by differences in nutrient availability (mainly glucose and cholesterol), favoring female fetuses with severe difficulties in development.

In this study, the negative effects of maternal undernutrition on the development of offspring in twin pregnancies were not modulated by the sex of the littermates, regardless of either concordant or discordant sex. An analysis of the potential differences between sex-discordant and sex-concordant twins showed that nutritional supplementation with concentrate feed increased the birth weight in both sexes, but this effect was higher in the case of a sex-discordant littermate; hence, both females and males with sex-discordant co-twins had a significantly higher birth weight than those with co-twins of the same sex. This effect was observed only for males in the case of antioxidant supplementation, which reinforces the impact of antioxidant supplementation on males. These results are in agreement with our previous studies in pigs. In the case of maternal undernutrition and antioxidant supplementation, males showed better indexes of antioxidant status and glucose metabolism, better activation of placental antioxidant defense genes, and, concomitantly, better compensatory growth patterns than undernourished males [48]. However, the greatest weight increase in newborn lambs, regardless of sex, was obtained when mothers were treated with a combination of nutritional supplements and antioxidants. Likewise, males were significantly heavier than females and no discordant sex differences were observed in either females or males. These results reinforce the role of both maternal undernutrition and litter size in diminished oxygen supply to the fetuses, resulting in decreased intrauterine growth [21]. However, the different results of each particular treatment suggest different responses to fetal growth according to the particular distribution of fetal sex in the twin pair and the maternal environment. Previous studies in human medicine have shown that, under adequate nutritional conditions, sex-discordant newborns have a higher total birth weight than twins of the same sex [49]. Two elegant retrospective analyses of 2491 and 67,850 twin pregnancies [50,51] showed better fetal growth in both males and females in same-sex twin pairs compared to discordant-sex male or female pairs. A possible explanation could be related to the finding that, in discordant-sex twin pairs, the female twin prolongs gestation for her brother, resulting in a higher birth weight for the male twin than that of same-sex male twins [52]; however, such an explanation may not apply since females of discordant-sex pairs have also been found to have higher birth weights than females of same-sex pairs [50]. The effect of the sex of the offspring observed in our study supports the aforementioned observations in human newborns, providing a better understanding of the maternal–fetal and feto–fetal relationships that affect intrauterine growth in harsh environments. In addition, these results support the robustness of the use of the sheep model in translational biomedical studies.

## 5. Conclusions

In the present study, we showed that in sheep twin pregnancies, chronic undernutrition abolished the male–female differences in body weight at birth. Both the restoration of the maternal nutritional plan, which was adjusted to the ewes’ requirements, and the supplementation of antioxidants tended to normalize birth weight and restore its differences between females and males, an effect that was enhanced with the combined supplementation of concentrated food and antioxidants. In each treatment, differences in newborns’ birth weight were expressed as a function of the concordance or discordance of sexes between the pairs of twins, suggesting different responses according to the particular maternal environment. The intrauterine physiological factors mediating these effects need to be clarified in future research.

## Figures and Tables

**Table 1 animals-14-00974-t001:** Experimental groups according to the diet offered daily during pregnancy.

	Total Diet		
Group	Base Diet	Concentrate Supplementation	Premix Antioxidant Supplementation
C	Grazing on natural prairie	50 g	None
N	Grazing on natural prairie	450 g	None
A	Grazing on natural prairie	None	50 g
NA	Grazing on natural prairie	400 g	50 g

Abbreviations: C = control undernourished group; N = group supplemented with concentrated food to cover 100% of requirements; A = group supplemented with antioxidants; NA = group supplemented with concentrated food to cover 100% of requirements plus antioxidants.

**Table 2 animals-14-00974-t002:** Maternal body weight (BW) and body condition score (BCS) at the beginning and at the end of pregnancy.

	BW (Kg)			BCS		
Group	Initial	Final	*p*-Value	Initial	Final	*p*-Value
C	62.0 ± 1.2 ^a^	61.1 ± 1.5 ^b^	ns	2.2 ± 0.1 ^a^	1.2 ± 0.1 ^b^	0.01
N	60.4 ± 1.1 ^a^	72.8 ± 1.0 ^a^	<0.001	2.0 ± 0.2 ^a^	1.9 ± 0.2 ^a^	ns
A	60.9 ± 1.6 ^a^	58.7 ± 1.1 ^b^	ns	2.5 ± 0.3 ^a^	1.0 ± 0.0 ^b^	<0.001
NA	60.5 ± 1.3 ^a^	70.2 ± 1.3 ^a^	<0.001	2.4 ± 0.2 ^a^	2.1 ± 0.2 ^a^	ns

Abbreviations: C = control undernourished group; N = group supplemented with concentrated food; A = group supplemented with antioxidants; NA = group supplemented with concentrated food and antioxidants. Different superscript letters in the same column indicate significant differences among groups (*p* ≤ 0.05).

**Table 3 animals-14-00974-t003:** Group distribution of twin-bearing ewes in accordance with their newborn sex pairs and comparison of average birth weight between the sexes.

	Sex Pairs (%)				Birthweight (kg)		
Group	n	FM	FF	MM	Females	Males	*p*-Value
C	23	52.2	21.7	26.1	3.34 ± 0.12 ^b^	3.38 ± 0.15 ^c^	ns
N	33	45.5	36.4	18.1	3.85 ± 0.10 ^a^	4.21 ± 0.10 ^ab^	0.015
A	20	43.7	31.3	25.0	3.65 ± 0.17 ^ab^	3.89 ± 0.10 ^b^	0.067
NA	20	50.0	30.0	20.0	4.01 ± 0.09 ^a^	4.56 ± 0.13 ^a^	0.038

Abbreviations: C = control undernourished group; N = group supplemented with concentrated food; A = group supplemented with antioxidants; NA = group supplemented with concentrated food and antioxidants. F = female; M = male. Different superscript letters indicate significant differences among the values in the same column (*p* < 0.05).

**Table 4 animals-14-00974-t004:** Effects of nutritional and/or antioxidant supplementation during gestation on lambs’ birth weight in accordance with their co-twin sex in undernourished twin sheep pregnancies.

	Lamb’s Birth Weight (Kg)				*p*-Value
Lamb Sex	C	N	A	NA	
F(M)	3.35 ± 0.15 ^b^	4.03 ± 0.26 ^a-2^	3.38 ± 0.10 ^b-3^	3.98 ± 0.10 ^a-2^	<0.001
F(F)	3.33 ± 0.18 ^c^	3.74 ± 0.13 ^bc-2^	3.83 ± 0.26 ^bc-2,3^	4.03 ± 0.14 ^ab-2,3^	0.059
M(F)	3.37 ± 0.24 ^b^	4.48 ± 0.13 ^a-1^	4.09 ± 0.18 ^a-1,2^	4.39 ± 0.13 ^a-1,3^	<0.001
M(M)	3.40 ± 0.17 ^b^	3.72 ± 0.01 ^b-2^	3.86 ± 0.15 ^b-2,3^	4.77 ± 0.26 ^a-1^	<0.001
***p*-value**	0.207	<0.001	0.077	0.003	

Abbreviations: F = female; M = male; (F) = female couple; (M) = male couple; C = control undernourished group; N = group supplemented with concentrate food; A = group supplemented with antioxidants; NA = group supplemented with concentrate food and antioxidants. Different superscript letters indicate significant differences among the values in the same row. Different superscript numbers indicate significant differences among the values in the same column.

## Data Availability

Data are contained within the article.

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
