# Peer review of "Chronic Undernutrition in Ovine Twin Pregnancies Abolishes Differences in Birth Weight Due to Sex: An Evaluation of the Role of Nutritional and Antioxidant Supplementation"

_animals, 2024, doi:10.3390/ani14060974_

Round 1

Reviewer 1 Report

Comments and Suggestions for Authors

The manuscript titled "Chronic undernutrition in ovine twin pregnancies abolishes differences in birth weight due to sex: evaluation of the role of nutritional and antioxidants supplementation" evaluate the effect of different nutritional approaches (using antioxidants and extra-supplementation) on the birth weight of twins in lamb. This study analyses variations in the body weight at birth, and it is hypothesized that they are associated to the fetus sex. The paper is well-written, although in some parts is difficult to follow. Some quaestions and suggestions are now listed:

In section 2.1 (L11-130), it is not clear the diet and supplement/treatment received by each experimental group. A table could be interesting for a better understanding. It is difficult to discern the differences between groups A and NA. Authors said that A group received a premix in a concentrate similar to the supplied to the N group; please, clarify the concentrate quantity administered to these groups.

L199: "21.0%" Comma should be replaced by dot.

Control group is considered as underfed group. However, ewes are rearing on a natural prairie and the can get enough nutrients. Final BW was not different from initial BW; then, did "underfed" produce "undernutrition" in C group ewes? How do you confirm undernutrition in the ewes? It is important to clarify this aspect.

Reviewer 2 Report

Comments and Suggestions for Authors

Dear authors,

Your experimental plan is based on the effect of undernutrition with supplementation for antioxidants in pregnant sheep to modulate the birth weight of twin lambs.

I believe that the experimental design needs additional refence if available (also from human medicine) to the effect of this combined nutritional and metabolic condition in sheep. The effect of the mother metabolism and diet regimen is well known to affect offspring digestive efficiency by up regulating specific enzyme synthesis for digestion.

However, I am afraid that some points still are not clear to me. 

I invite the authors to provide a sufficient background based on reference that led them to hypothesize why antioxidant and malnutrition could have a linear effect on modulating the birth weigh of twin lambs. At this regard, I believe that a similar paper in goats showed how different metabolic condition and body fluid distribution in the mother could be seen in single vs. twin gestation and the differences also obesrved on birth weigh of singleton vs twins. Probably such information would sustain evidence for the formulated hypothesis.

The sex ratio and the differences in weight need a better argumentation. When you have twins of the same sex, it can occur that one out of the two weigh more than the other. 

Probably, not only weight should be screened but also morphometry of male vs. female.

This is my suggestion to improve the paper. 

Finally, I do believe that the text needs to be proofread to better use technical terms for the description of the background, experimental activities and discussion of results.

Best regards.

Comments on the Quality of English Language

The paper needs extensive editing of language, not only for grammar amendments, but also for technical terms that need to be better used for allowing the reader catch the correctness of experimental plan.

Reviewer 3 Report

Comments and Suggestions for Authors

Thank you for the article sent, however, I have a few questions about it

- there is no information what breed the sheep participating in the experiment are, in order to have some reference for the results obtained

- the BCS result from the end of pregnancy from line 106 does not correspond to the result shown in Table 1. 

- in the methodology, you can also add a table that shows information on maternal requirements, and how the different diets cover this requirement. Here I also have a question about whether the amount of concentrate intake changed during pregnancy, since the demand at the beginning and end of pregnancy is quite different

- Table 2.- what is the reason for the unequal distribution between the different groups 

- it would also be good to state, in addition to the % of individual pairs of twins, how many lambs there were in each group (it seems to me that there may be quite a large numerical variation of these lambs here), this will also allow a better interpretation of the results

- lines 193, 199, 202, 204 and 207 you can add that this increase occurred compared to the control group